Ultra-processed food consumption is linked to quality of life and mental distress among university students

Ertaş Öztürk Yasemin yasemnertas@gmail.com
Uzdil Zeynep
Department of Nutrition and Dietetics, Faculty of Health Sciences, Ondokuz Mayıs University , Samsun , Türkiye
Myers Catherine
Electronic publication date: 2025 Aug 25
Publication date: 2025
Volume: 13
Electronic Location ID: e19931
Received 2024 Oct 23; Accepted 2025 Jul 25
Copyright: © 2025 Ertaş Öztürk and Uzdil
Copyright year: 2025
Copyright holder: Ertaş Öztürk and Uzdil
License: This is an open access article distributed under the terms of the Creative Commons Attribution License, which permits unrestricted use, distribution, reproduction and adaptation in any medium and for any purpose provided that it is properly attributed. For attribution, the original author(s), title, publication source (PeerJ) and either DOI or URL of the article must be cited.
License URL: https://creativecommons.org/licenses/by/4.0/

Keywords: Mood, Dietary pattern, Ultra-processed foods, Quality of life, Mental distress

Funding: Ondokuz Mayıs University Scientific Research Projects Coordination Unit PYO.SBF.1902.22.001 This work was supported by “Ondokuz Mayıs University Scientific Research Projects Coordination Unit with the project number: PYO.SBF.1902.22.001.” The funders had no role in study design, data collection and analysis, decision to publish, or preparation of the manuscript.

==============================
Background

Ultra-processed foods (UPFs) are industrial formulations that typically contain little or no whole foods and are often high in added sugars, unhealthy fats, and sodium. Research indicates that higher intake of UPFs correlates with lower overall diet quality, which can exacerbate mental health issues such as anxiety and depression. This study aimed to assess the associations between UPF consumption with health-related quality of life, and mental distress in university students.

Method

This cross-sectional study consisted of 595 students resident in Samsun/Türkiye. The data were collected face-to-face with a questionnaire. Linear regression analysis was conducted to examine the relationship between the UPFs (% energy), health-related quality of life components and mental distress score (MDS). Receiver operating characteristic (ROC) analysis was conducted to establish the optimal threshold levels regarding physical component summary (PCS12), mental component summary (MCS12) and MDS.

Results

The UPF ratios were negatively related to PCS12 (β = −0.112, p = 0.005) and positively associated with MDS (β = 0.102, p = 0.002) after adjusting for age, sex, physical activity, smoking duration, number of cigarettes and chronic disease. ROC analysis showed that those with UPF consumption above 45.096% had low physical quality of life, those above 50.999% had low mental quality of life, and those above 40.250% had moderate-to-high mental distress.

Conclusion

Increased consumption of UPFs is associated with a decrease in physical quality of life and an increase in mental distress. Consuming more than 40% from UPFs can lead to mental and physical problems. There is a need for strategies to decrease the intake of UPFs to enhance both physical and mental wellbeing.

Introduction

From young to adulthood, university acts as a transitional phase that helps to develop abilities essential for future careers (Uzdil, Kılıç & Özenoğlu, 2021). This can be difficult due to several reasons, including peer competition, academic performance pressure, modifications in workload and support networks, changes in living conditions, and extended separation from family (Ramón-Arbués et al., 2022). The incidence of mental distress indicators, including depression and anxiety, among university students ranges from 18% to 50% (Mutinta, 2022; Ramón-Arbués et al., 2022). Mental health impairment is more pronounced among university students, and both physical and mental health impairments are inversely correlated with health-related academic success (Wilks et al., 2020).

University students may have limited access to healthy, home-cooked meals and may depend on readily available, quickly consumable items from vending machines. Additionally, university canteens predominantly offer energy-dense snacks (Bos et al., 2018; Racine et al., 2022). Because of budget limitations and prolonged study periods, students frequently lack the opportunity to cook their own meals (Durán-Agüero et al., 2023). However, food preferences and dietary choices play an essential role in promoting physical and mental health (Mikkawi, Khoury & Rizk, 2024). From the early years of life, proteins, polyunsaturated fatty acids (arachidonic acid, docosahexaenoic acid, eicosapentaenoic acid, and gamma-linolenic acid), B vitamins (such as B1, B6, and B12), and minerals (e.g., selenium, zinc, magnesium, iron, copper, and iodine) play crucial roles in the development and healthy functioning of the nervous system (Wendołowicz, Stefańska & Ostrowska, 2018). The Mediterranean diet is abundant in grains, fruits, vegetables, legumes, nuts, oilseeds, and olives; high to moderate in fish and seafood; moderate in eggs, poultry, and dairy; low in red meat; and is composed of nutritional elements advantageous for the nervous system (Bach-Faig et al., 2011). Adherence to high-quality diets like the Mediterranean diet correlated with reduced depression, anxiety, and stress-related mental quality of life in a cross-sectional study of female adults (Açik, Altan & Çakiroğlu, 2022). In parallel, western diets high in fats and processed foods adversely impact mental health (Aslan Çin et al., 2022). Like a vicious circle, adverse emotional states promote high-fat and sugary food intakes (Fong et al., 2019). The dietary inflammatory burden was observed to be elevated in female university students with moderate to severe depression (Açik & Çakiroğlu, 2019).

The NOVA categorization system categorizes foods according to their level of natural content and the extent of industrial processing they undergo. It categorizes foods into four classifications: unprocessed and minimally processed foods (UPMP), processed culinary ingredients (PCI), processed foods (PF), and ultra-processed foods (UPF) (Monteiro et al., 2019). UPMP includes raw or minimally processed fresh fruits, vegetables, and meats. PCI includes oils, sugars, and salts used in culinary practices. PF includes foods that are processed, including canned vegetables or fruits with added salt or sugar. Lastly, UPF is characterized as “industrial preparations that are ready-to-eat or ready-to-heat, primarily composed of substances derived from food, frequently chemically altered, containing additives, and comprising a minimal amount of whole food” (Monteiro et al., 2019; Vitale et al., 2024). UPFs possess high levels of fat, salt, or sugar and are typically deficient in dietary fiber, protein, micronutrients, and beneficial functional substances (Monteiro et al., 2019). It has attracted attention due to its relation with multiple health effects (Pagliai et al., 2021; Juul et al., 2022).

Quality of life has been defined as “an overall general well-being that comprises objective descriptors and subjective evaluations of physical, material, social, and emotional well-being together with the extent of personal development and purposeful activity, all weighted by a personal set of values” (Karimi & Brazier, 2016). Recent studies indicate that the intake of UPFs significantly increases the risk of chronic diseases, including obesity, hypertension, cardiovascular diseases, type 2 diabetes mellitus, and cancer (Pagliai et al., 2021; Juul et al., 2022; Delpino et al., 2022). Furthermore, individuals who consume a substantial proportion of UPFs (51.1% of energy compared to 12.4% of energy) exhibit a deterioration in health related quality of life (HRQoL) (Cadenhead et al., 2023).

Mental distress is defined as “a significant disturbance in emotional processes, thoughts, or cognitions that impairs judgment, behavior, or the ability to cope with the ordinary demands of life” (Mutinta, 2022). A longitudinal study (Gómez-Donoso et al., 2020) of young adults possessing a college degree indicated that increased consumption of UPFs was associated with an elevated risk of depression. A similar finding was observed in a cohort study involving adults and the elderly (Adjibade et al., 2019). The consumption of UMPF correlates with reduced levels of depression. In contrast, the consumption of UPFs elevates the risk of anxiety (Coletro et al., 2022). A meta-analysis of 17 non-interventional observational studies indicated that the consumption of UPF was correlated with anxiety and depression (Lane et al., 2022a). A longitudinal study indicates that elevated consumption of UPFs correlates with depression in healthy adults (Ferreira et al., 2024).

Although the associations between UPF consumption and HRQoL or mood have primarily been shown in adults and the elderly, there is limited research examining this relationship among university students. This study aimed to assess the correlations between UPF intake, HRQoL, and mood among university students. Moreover, our aim was to set a potential cut-off above which UPF intake diminishes quality of life and exacerbates mental distress.

Material and method

Study design and subjects

This cross-sectional study was conducted on 623 university students aged 18–25 years at Ondokuz Mayıs University Kurupelit Campus between October 2022 and June 2023 in Samsun/Türkiye. The sample size calculation was calculated to be 580 with 95% CI and 5% margin of error, based on the prevalence of consuming UPF ≥5 times per week, reported as 78.2% in the Spanish adolescent population (Ruíz-Roso et al., 2020). To minimize potential selection bias in participant recruitment, the study was conducted in multiple departments across the university campus and a central campus hub that includes food courts, cafes, a supermarket, stationery, game areas. Any university students who were not pregnant or breastfeeding were included. After excluding those with energy intakes fewer than 800 and greater than 4,000 kcal (Banna et al., 2017), the study was completed with 595 students.

Ethical permission was obtained from Ondokuz Mayis University, Clinical Research Ethics Committee (2022/73-B.30.2.ODM.0.20.08/87 and date: 10.02.2022). Written consent of participants was taken for their volunteer participation and the principles of the Helsinki Declaration were applied.

Data collection and ethic statements

Data collection from volunteer participants was conducted through face-to-face administration of a questionnaire prepared by the researchers. The questionnaire form included demographic information, the Food-Mood Questionnaire, the Health-Related Quality of Life Short Form-12, and a Food Frequency Questionnaire (FFQ) categorized according to the NOVA classification system. Additionally, participants’ height was measured using a portable stadiometer, and body weight was assessed using a portable bioelectric impedance analysis device. Subsequently, the body mass index (BMI) was calculated using the recorded height and weight values.

Instruments

Health-related quality of life

The short form (SF)-12 developed by Ware, Kosinski & Keller (1998) was used to evaluate health-related quality of life. The Turkish reliability and validity study was conducted by Soylu & Kütük (2022). Similar to the widely used Quality of Life Short Form 36 (SF-36), it consists of eight factors and 12 items: physical functioning (two items), role-physical (two items), bodily pain (one item), general health (one item), energy/fatigue (one item), social functioning (one item), role-emotional (two items) and mental health (two items). Items regarding physical and emotional roles were answered yes or no; other items have Likert-type responses ranging from 3 to 6 points. The total health-related quality of life score is evaluated separately into two summary scores: the physical component summary (PCS12) and the mental component summary (MCS12). The PCS12 is obtained from the factors of general health, role-physical, physical functioning, and bodily pain, and the MCS12 score is obtained from the factors of social functioning, role-emotional, mental health, and energy/fatigue. Both scores range from 0 to 100, with a higher score representing better health. In the present study the Cronbach alpha values were 0.712 for PCS12 and 0.742 for MCS12.

Mental distress

The mood section of the Food-Mood Questionnaire was used in determining the Mental Distress Score (MDS). The Food-Mood Questionnaire was developed by Begdache, Marhaba & Chaar (2019) to measure the effect of foods on mental distress. The reliability and validity of the Turkish version of the scale were determined by Aslan Çin et al. (2022). The scale is a six-point Likert type and asks for answers ranging from 0 (never) to 6 (more than four times) and, has 21 items and consists of five factors, including mental distress pattern (six items), healthy pattern (six items), breakfast pattern (four items), Western-type pattern (three items), and nutritional supplement pattern (two items). In the present study the Cronbach’s alpha value was 0.840 for MDS.

Consumption of food groups according to NOVA

To determine the consumption of food groups according to NOVA of the volunteers, a FFQ based on the foods described in the FAO report of the NOVA classification (Monteiro et al., 2019), with consideration given to the foods often consumed in Turkish culture was applied. The data were evaluated through the Turkish Nutrition Information System program (BeBiS, 9.1), and the energy intakes of individuals were calculated. The energy proportions (%E) from UMPF, PCI, PF, and UPF were used in the statistical analysis.

Statistical analysis

Numeric data are presented as mean and standard deviation (SD) and categorical data are presented as numbers and percentages (%). The normality of the variables was assessed using the kurtosis and skewness coefficients. Both coefficients, which ranged between (−2, 2) confirm that the normality assumption was met. Pearson correlation analysis was used to assess relationships between consumption of NOVA groups (UMPF, PCI, PF, and UPF) and PCS12, MCS12 and MDS to assess the relationships. The significant relationships identified in the correlation analysis were further included in the linear regression models. Two adjusted models (Modela: based on age, gender, physical activity and Modelb: age, gender, physical activity, smoking duration, number of cigarettes and chronic disease) were performed to measure the effects of UPF consumption on PCS12, MCS12 and MDS. A stepwise procedure was performed to eliminate insignificant factors from the models. Finally, we conducted receiver operating characteristic curve (ROC) analysis after the regression models to find clinically useful cut-off points for UPF consumption. This method is especially useful in nutritional epidemiology, where continuous dietary measures need to be turned into public health recommendations. This method has let us turn the amount of UPF consumption, which is a continuous variable, into a classification tool. This approach can be applied to all data containing continuous data that can be categorized. The relevance of ROC analysis in similar contexts has been demonstrated in the literature (Rankinen et al., 1999; Mirmiran, Esmaillzadeh & Azizi, 2004). A few examples are as follows, Libardi et al. (2025) determined the cut-off point for neck circumference using ROC analysis in their study. Nakamura et al. (2024) used ROC analysis to determine the BMI value associated with increased insulin use in GDM. After the literature review, we did not find any cut-off points where UPF consumption decreased quality of life or increased mental distress.

The ROC curve is constructed by plotting the true positive rate (sensitivity) against the false positive rate (1-specificity) across a range of possible cut-off points (Unal, 2017). The Youden index was used to identify optimal cut-offs as it balances sensitivity and specificity, providing the most clinically useful threshold. In this study, we aimed to conduct further analysis to find the point at which UPF (E%) intake decreased HRQoL and increased mental distress. By doing so, we determined the value of the UPF (E%) intakes that best show the difference between participants with higher vs lower HRQoL or low vs moderate-to-high mental distress. For that, the PCS12 and MCS12 were divided into two categorical groups using standardized median values of the component summaries as ≥50 points (accepted as a reflection of high physical or mental quality of life) and <50 points (accepted as a reflection of low physical or mental quality of life). The MDS was divided into ≥6 points and <6 points to reflect moderate-to-high mental distress (Begdache, Marhaba & Chaar, 2019). All the statistical findings were obtained using IBM SPSS 28 (IBM Corp., Armonk, NY, USA) and R software, and the type-1 error level was set at α = 0.05.

Results

General information about the participants is given in Table 1. This study reached 595 individuals, 195 men (32.8%) and 400 women (67.2%), with a mean age of 21.2 ± 1.45 years.

Table 1 Characteristics of individuals.

Characteristics	Mean/N	SD/%	
Age (yr)	21.2	1.45	
Sex (women, %)	400	67.2	
Body mass index (kg/m2)	22.9	3.70	
Marital status (single, %)	593	99.7	
Living with family (no, %)	464	78.0	
Working status (no, %)	559	93.9	
Smoking status (yes, %)	151	25.4	
Duration of smoking (yr)	4.1	3.09	
Number of smoke (t/d)	11.9	6.72	
Alcohol use (yes, %)	111	18.7	
Chronic disease (yes, %)	52	8.7	
Physical activity level (%)			
Low (≤1 t/w)	314	52.8	
Moderate (2–3 t/w)	160	26.9	
High (≥4 t/w)	121	20.3	
Health-related quality of life (SF-12)			
Physical component summary	52.2	7.01	
High (≥50 point (%))	425	71.4	
Mental component summary	38.9	10.34	
High (≥50 point (%))	98	16.5	
Mental distress score	11.9	5.15	
Moderate-to-high (≥6 point (%))	530	89.1	
NOVA food groups (energy, %)			
Unprocessed and minimally processed foods	31.2	10.81	
Processed culinary ingredients	9.74	5.67	
Processed foods	15.1	7.53	
Ultra-processed foods	43.9	14.35	

Figure 1 shows the correlation results between the NOVA food group consumptions (energy %) and PCS12, MCS12 and MDS. We observe some relevant associations between PCS12, MCS12, MDS and UMPF, PCI, PF, and UPF. UPF were negatively associated with UPMP (r = −0.745, p < 0.001), PCI (r = −0.414, p < 0.001), PF (r = −0.524, p < 0.001), PCS12 (r = −0.211, p < 0.001) and MCS12 (r = −0.093, p = 0.024) while UPF were positively associated with MDS (r = 0.179, p < 0.001). UPMP were positively associated with PCS12 (r = 0.179, p < 0.001) and MCS12 (r = 0.083, p = 0.043). MDS was negatively associated with MCS12 (r = −0.608, p < 0.001). PF was not associated with any variables.

Figure 1 Correlation results between the measurements and scale scores.

UPMP, unprocessed and minimally processed food; PF, processed food; PCI, processed culinary ingredient; UPF, ultra-processed food; MDS, mental distress score; PCS12, physical component; MCS12, mental component. The symbol X points out the insignificance of the correlation coefficient.

A linear regression analysis was conducted for both unadjusted and adjusted models for each dependent variable (PCS12, MCS12, and MDS). The stepwise procedure eliminated the other independent variables (UMPF, PCI, PF), and in the final model, only UPF is statistically significant. In Table 2, the results showed a substantial impact of UPF on PCS12, MCS12, and MDS in unadjusted model. However, UPF does not have a significant effect on MCS12 after adjusting the models. After adjustment for age, sex, physical activity, smoking duration, number of cigarettes and chronic disease, UPF was negatively associated with PCS12 (β = −0.112, p = 0.005) and positively related to MDS (β = 0.102, p = 0.002).

Table 2 Results of the regression models.

Dependent variable	Model	Coefficient	B	SB	t	p	95% Confidence interval	R 2	Adj- R2	
Lower bound	Upper bound	
PCS12	Unadjusted model	(Constant)	56.714		62.579	<0.001	54.934	58.494	0.045	0.043	
UPF	−0.103	−0.211	−5.256	<0.001	−0.142	−0.065	
Adjusted modela	(Constant)	59.470		13.524	<0.001	50.833	68.106	0.084	0.077	
UPF	−0.085	−0.174	−4.307	<0.001	−0.124	−0.046	
Age	0.011	0.002	0.058	0.954	−0.365	0.387	
Gender (Women)	−2.528	−0.169	−4.199	<0.001	−3.710	−1.345	
Physical activity (Moderate)	−0.813	−0.051	−1.241	0.215	−2.098	0.473	
Physical activity (High)	1.655	0.095	2.252	0.025	0.212	3.097	
Adjusted modelb	(Constant)	44.062		5.084	<0.001	26.932	61.193	0.264	0.228	
UPF	−0.112	−0.217	−2.823	0.005	−0.190	−0.033	
Age	0.208	0.043	0.588	0.558	−0.493	0.910	
Gender (Women)	−3.160	−0.220	−2.839	0.005	−5.360	−0.959	
Physical activity (Moderate)	1.087	0.067	0.882	0.379	−1.350	3.523	
Physical activity (High)	1.918	0.096	1.232	0.220	−1.160	4.995	
Smoking duration	−0.187	−0.081	−1.037	0.302	−0.543	0.169	
Number of cigarettes	−0.159	−0.148	−1.884	0.062	−0.326	0.008	
Chronic disease (No)	7.705	0.301	4.103	<0.001	3.993	11.416	
MCS12	Unadjusted model	(Constant)	41.808		30.713	<0.001	39.134	44.481	0.009	0.007	
UPF	−0.067	−0.093	−2.266	0.024	−0.125	−0.009	
Adjusted modela	(Constant)	34.339		5.118	<0.001	21.162	47.517	0.019	0.012	
UPF	−0.051	−0.070	−1.683	0.093	−0.110	0.008	
Age	0.332	0.046	1.132	0.258	−0.244	0.909	
Gender (Women)	−1.189	−0.054	−1.288	0.198	−3.003	0.624	
Physical activity (Moderate)	1.491	0.064	1.485	0.138	−0.481	3.463	
Physical activity (High)	1.572	0.061	1.396	0.163	−0.640	3.785	
Adjusted modelb	(Constant)	36.338		2.555	0.012	8.228	64.447	0.021	−0.027	
UPF	−0.014	−0.020	−0.221	0.826	−0.143	0.114	
Age	0.208	0.030	0.358	0.721	−0.940	1.356	
Gender (Women)	0.392	0.019	0.215	0.830	−3.212	3.996	
Physical activity (Moderate)	−0.131	−0.006	−0.065	0.949	−4.121	3.860	
Physical activity (High)	2.818	0.099	1.105	0.271	−2.223	7.858	
Smoking duration	−0.217	−0.066	−0.736	0.463	−0.800	0.366	
Number of cigarettes	0.156	0.102	1.126	0.262	−0.118	0.429	
Chronic disease (No)	−2.396	−0.066	−0.779	0.437	−8.475	3.683	
MDS	Unadjusted model	(Constant)	9.035		13.496	<0.001	7.720	10.350	0.032	0.030	
UPF	0.064	0.179	4.434	<0.001	0.036	0.093	
Adjusted modela	(Constant)	9.493		2.871	0.004	3.000	15.987	0.038	0.032	
UPF	0.058	0.163	3.931	<0.001	0.029	0.087	
Age	−0.043	−0.012	−0.301	0.764	−0.327	0.240	
Gender (Women)	0.831	0.076	1.832	0.067	−0.060	1.722	
Physical activity (Moderate)	−1.035	−0.089	−2.097	0.036	−2.004	−0.066	
Physical activity (High)	−0.137	−0.011	−0.248	0.804	−1.224	0.950	
Adjusted modelb	(Constant)	8.275		1.149	0.253	−5.962	22.511	0.080	0.035	
UPF	0.102	0.268	3.116	0.002	0.037	0.167	
Age	−0.028	−0.008	−0.094	0.925	−0.610	0.555	
Gender (Women)	0.208	0.020	0.225	0.823	−1.620	2.035	
Physical activity (Moderate)	0.493	0.041	0.482	0.631	−1.530	2.517	
Physical activity (High)	0.033	0.002	0.025	0.980	−2.523	2.589	
Smoking duration	0.233	0.135	1.557	0.122	−0.063	0.528	
Number of cigarettes	−0.046	−0.058	−0.655	0.513	−0.185	0.093	
Chronic disease (No)	−0.698	−0.037	−0.448	0.655	−3.781	2.384	
Notes:

PCS, Physical component summary; MCS, mental component summary; MDS, mental distress score; B, beta coefficient; SB, standardized beta coefficient.

a Adjusted based on age, sex, physical activity.

b Adjusted based on age, sex, physical activity, smoking duration, number of cigarettes and chronic disease.

Adj- R2, Adjusted determination coefficient. Statistically significant p values are shown in bold.

The results of the ROC analysis are presented in Table 3, and the corresponding ROC curves can be found in Fig. 2. Upon careful examination of the area under the curve values, it becomes evident that all of them significantly surpass the 0.5 threshold, providing sufficient evidence for substantial discrimination between the groups. The ROC analysis revealed statistically significant but modest discrimination ability for all three measures, with AUC values ranging from 0.574 to 0.642. While these values fall below the conventional threshold of 0.7 for good predictive performance, they indicate discrimination better than chance (Carrington et al., 2022). The optimal cut-off values for UPF consumption are 45.096% for the PCS12 group (high vs low physical quality of life), 50.999% for the MCS12 group (high vs low mental quality of life), and 40.250% for the MDS group (low vs moderate-to-high mental distress). In other words, it was found that those with UPF consumption above 45.096% had low physical quality of life, those above 50.999% had low mental quality of life and those above 40.250% had moderate-to-high mental distress in this study.

Table 3 ROC analysis results for three groups.

Group	AUC	SE	p	Sensitivity	Specificity	95% Confidence interval	Cut-off*	
Lower	Upper	
PCS12	0.614	0.026	<0.001	0.624	0.584	0.563	0.665	45.096	
MCS12	0.574	0.031	0.015	0.334	0.816	0.515	0.634	50.999	
MDS	0.642	0.036	<0.001	0.642	0.600	0.571	0.712	40.250	
Note:

* Optimal cut-off was found based on Youden index. The groups were created from the midpoints of the PCS12, MCS12 and MDS scores.

Figure 2 ROC curves for the outcomes.

(A) ROC curve for PCS12, (B) ROC curve for MCS12, (C) ROC curve for MDS.

Discussion

Within this cross-sectional study of university students, we showed associations of UPF consumption with quality of life and mental distress. We established threshold values of UPF according to physical, mental quality of life components and mental distress.

One of the main results of this study was that both physical and mental quality of life components were negatively associated with UPF intake in correlation (Fig. 1) and unadjusted linear regression model (Table 2). After the adjustments of age, sex and physical activity the associations were still significant (Table 2). Health-related quality of life is based on an individual’s subjective evaluation of his own health and wellbeing and is a valuable outcome of healthcare (Vajdi & Farhangi, 2020). In a longitudinal study, there was a notable decrease in the quality of life ratings, partially ascribed to the rise in several adjustable risk factors (Olfson et al., 2018). There are a limited number of studies showing associations between UPF consumption and quality of life. Our findings relating to the UPF consumption and quality of life are reported for the first time with this study in Türkiye and are in agreement with recent limited literature. In a longitudinal study, unhealthy dietary habits such as UPF and sweet intakes were associated with lower health related quality of life in colorectal cancer survivors (Kenkhuis et al., 2023). In another study, Iranian adolescents with higher UPF consumption had poorer quality of life (Lane et al., 2022b). These findings are parallel with the results showed associations between “unhealthy” or “Western diet” and depressive symptoms (Vajdi & Farhangi, 2020) or adherence to an MD pattern is associated with better quality of life (Bonaccio et al., 2013).

We further determined a cut-off and found the UPF intake above 45.096% for physical component and 50.999% for the mental component of SF-12 was optimal to show the decreasing quality of life of each representative component (Fig. 2, Table 2). Zhang & Giovannucci (2023) reported that younger individuals tended to consume more UPFs. Moreover, elevated UPF consumption correlated with younger age, urban residency, and being single, such as unmarried, separated or divorced. Education, income, and social status had differing correlations according to the country (Dicken, Qamar & Batterham, 2023). The consumption of UPFs is linked to various health outcomes, including heightened risk of all-cause mortality, and noncommunicable diseases such as cardiovascular diseases, cerebrovascular diseases, hypertension, metabolic syndrome, obesity, depression, irritable bowel syndrome, cancers, gestational obesity, asthma and wheezing, and frailty, as indicated by cohort and cross-sectional studies (Chen et al., 2020). In a meta-analysis, increased UPF consumption was associated with a worse cardiometabolic risk profile and a higher risk of cardiovascular disease, cerebrovascular disease, depression and all-cause mortality (Pagliai et al., 2021). Similarly, Juul et al. (2022) concluded that a greater overall consumption of UPFs correlates with an elevated risk of type 2 diabetes, obesity, and cardiovascular disease in adulthood. UPF consumption is related to diabetes risk in a dose-dependent manner (Delpino et al., 2022) and higher consumption of UPF was strongly associated with a higher risk of multiple indicators of obesity in the adult population over a 13-year follow up (Rauber et al., 2021). These findings indicate that the percentage of energy from UPF consumption, which we have determined in our study, should be reduced. By reducing UPF consumption, the quality of life can be improved through disease prevention.

In this study, we demonstrated a positive association between UPF intake and mental distress (Fig. 1) and the associations were still significant after the adjusting for possible confounders (Table 2). Although the associations between diet and mental health are complex, diet can have direct effects on mood (Bremner et al., 2020). Previous investigations based on dietary components or patterns were previously related to mental health. They showed that diet has a fundamental role by being essential for the brain, having a role in the regulation of circadian rhythm, and the gut-brain axis, and contributing to inflammation and oxidative stress (Godos et al., 2020). Adopting healthy eating habits was linked to reduced mental distress and increased psychological well-being (Hong & Peltzer, 2017) whereas a growing body of evidence showed that high fast-food consumption (Begdache et al., 2020) and abnormal calorie intake (Cheon et al., 2020) were related to mental distress. In a nationally representative sample of American adults, those who reported consuming larger amounts of UPFs were more prone to experiencing mild depression, having more days with mental illness and anxiety, and less likely to report having no mentally unhealthy or anxious days (Hecht et al., 2022). In another study which included 1,693 adults from Brazil, the presence of a combination of two or three health risk behaviours, such as consuming high amounts of UPFs and not consuming fruits and vegetables on a daily basis, resulted in a greater occurrence of symptoms associated with anxiety or depression (Coletro et al., 2022). Similarly, Lane et al. (2022a) reported that greater UPF consumption was cross-sectionally associated with increased odds of depressive and anxiety symptoms. In the NutriNet-Santé cohort which included 20,380 women and 6,350 men aged 18–86 years, UPF consumption was positively associated with the risk of incident depressive symptoms after follow-up of 5.4 years (Adjibade et al., 2019). A positive association between UPF consumption and likelihood of having depressive symptoms was found in younger southern Italian adults (Godos et al., 2023). Our results support the previous findings.

When we performed a ROC analysis, an energy intake above 40.250% from UPF was significantly associated with increased moderate to high mental distress among the participants (Fig. 2, Table 3). A limited number of studies have indicated the amount or energy ratio of UPF consumption in terms of mental distress. In the SUN project which included 14,907 Spanish university graduates and followed up for a median of 10.3 years, a total of 774 incident cases of depression were identified and a higher risk of developing depression was found in the highest UPF consumption quartile. However, in this study UPF amounts were represented as g/day and reported that the 4th quartile (192.7 g/day) was significantly different from the 1st quartile (51.8 g/day) (Gómez-Donoso et al., 2020). In a recent meta-analysis UPF consumption is related to an enhanced depressive mental health status risk in a dose-dependent manner. Additionally, a 10% increase in UPF consumption per daily calorie intake increased the depression risk by 11% (Mazloomi et al., 2023).

There are several possible mechanisms for UPF consumption that potentially explain their effects on increased chronic disease risk (Crimarco, Landry & Gardner, 2022), mental health and eventually health-related quality of life. Diets high in UPFs generally reflect poor diet quality (Marchese et al., 2022; Moubarac et al., 2017; Shim et al., 2022; Vandevijvere et al., 2019; Houshialsadat et al., 2024). First mention of UPFs, by Monteiro et al. (2011) as “ultra-processed ready-to-eat or ready-to-heat food products” were shown to be consumed by both lower and upper-income groups and characterized by more added sugar, more saturated fat, more sodium, less fibre and much higher energy density. In a meta-analysis of nationally representative samples of various countries, UPFs were consistently associated with worsening diet quality (Martini et al., 2021). In a narrative review by Elizabeth et al. (2020), other possible mechanisms reviewed included a higher glycaemic load of the UPFs, diminished satiety signalling due to processing, harmful components formed during high-temperature cooking, and pro-inflammatory responses associated with industrial food additives, and imbalances in gut microbial community. Moreover, in recent years UPFs have been associated with food addiction (LaFata & Gearhardt, 2022). Young adults (18–35 years) with food addiction were shown to have higher energy % from UPFs and lower energy % from UPMP in an Australian study (Whatnall et al., 2022). Our research revealed that the consumption of UPFs accounted for 43.9% of the total energy intake among university students. Our findings align with the existing research, which has indicated a global increase in the intake of UPFs. Recent studies have indicated that such foods contribute to a substantial proportion, approximately 50–60%, of the total energy content in the typical diet of the average consumer in the United States, Canada, or the United Kingdom (Pagliai et al., 2021). Beyond these countries, data from Korea, Mexico, Brazil, and Sweden have the same increasing consumption trend (Zhang & Giovannucci, 2023). Although, the traditional Turkish dietary patterns are based on the Mediterranean diet, characterized by high consumption of legumes, vegetables, fruits, whole grains, and olive oil, and relatively low intake of processed foods (Pekcan, 2018), a transition in the dietary patterns especially in young groups was reported (Akman et al., 2010). In Türkiye, there are limited studies on UPF consumption. However, a recent study with 2,512 university students reported the UPF consumption ratio as 69.3% (Eroğlu, Ekici & Göktürk, 2025). Unfortunately, the large availability of UPFs on the market is supporting this trend (Şimşek et al., 2025).

As far as we know, this is the first study in Turkey that looks at the link between UPF and mental distress and quality of life using ROC-derived threshold values. This study makes a new contribution to the field by suggesting specific cut-off points for UPF intake based on mental and physical health quality. In most cases, previous studies looked at the general links between UPFs and psychological outcomes such as depression and anxiety (Mazloomi et al., 2023; Hecht et al., 2022; Coletro et al., 2022). Our results, on the other hand, give a more complete picture by looking at mental distress, quality of life, and population-specific intake thresholds. This makes it easier to use the information to improve public health, especially in university students in whom UPF consumption is rising (Pagliai et al., 2021; Zhang & Giovannucci, 2023).

There are several strengths and limitations in this study. Relatively large sample size and face-to face data collection are among the strengths. Furthermore, the FFQ was designed according to NOVA, and was much less time consuming. Our preliminary results showed consistent results among food groups of the food-mood questionnaire and the FFQ (Ertaş Öztürk & Uzdil, 2024). We controlled our results according to the potential confounders based on their theoretical relevance, empirical evidence from previous research. However, other potential confounders, such as socioeconomic status, academic stress, and sleep quality, that were not considered, may be included in further studies. Another limitation is that we could not show any causality because of the nature of the cross-sectional study design. Also, the cross-sectional study design may limit the generalizability of the results to nationwide. Lastly, our results showed very high mental distress among this young group that can be a limitation. The results of this study should be replicated by further longitudinal studies, and the cut-off values we showed in our study should be justified.

Conclusion

Our findings indicate that higher UPF consumption is associated with lower quality of life and higher mental distress. Intakes above 40% of energy from UPF may lead moderate to high mental distress, while 45% may decrease physical, and 51% may decrease the mental quality of life. These results have important implications for public health policies and university-based interventions as they demonstrate links between UPF and a lower quality of life, increased mental distress, as they suggest threshold values. Consequently, these results carry significant implications for public health, particularly for policymakers dealing with the food industry who should oppose the intake of UPFs and advocate for the consumption of minimally processed foods. There is a need for radical strategies to reduce the consumption of UPFs for the promotion of physical and mental quality of life and prevention of mental distress. From a public health perspective, these results indicate the necessity for dietary guidelines that recommend limits on UPF. Establishing nutrition education programs, limiting the availability of UPFs on campus, and making it simpler for students to obtain affordable, minimally processed food options should be encouraged. At the policy level, government and institutional parties should consider front-of-package labelling indicating the degree of processing. Implementing preventive nutritional changes is crucial for safeguarding these vulnerable populations. These changes toward nutrition, with targeted preventative interventions in academic settings, have the potential to greatly improve young adults’ physical and mental health.

Supplemental Information

Supplemental Information 1 Raw data.

We would like to thank our students and all participants who supported the data collection phase of the project. We acknowledge to AI-based tool (PoolText) to provide checking and improving the grammar and the language clarity of the manuscript.

Additional Information and Declarations

Competing Interests

The authors declare that they have no competing interests.

Author Contributions

Yasemin Ertaş Öztürk conceived and designed the experiments, analyzed the data, prepared figures and/or tables, authored or reviewed drafts of the article, and approved the final draft.

Zeynep Uzdil conceived and designed the experiments, analyzed the data, prepared figures and/or tables, authored or reviewed drafts of the article, and approved the final draft.

Human Ethics

The following information was supplied relating to ethical approvals (i.e., approving body and any reference numbers):

The study was approved by the ethics committee of Ondokuz Mayıs University Clinical Research Ethics Committee (decision number: 2022/73- B.30.2.ODM.0.20.08/87 and date: 10th of February 2022).

Data Availability

The following information was supplied regarding data availability:

Raw data is available in the Supplemental Files.

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
