# Peer review of "Ultra-processed food consumption is linked to quality of life and mental distress among university students"

_PeerJ, doi:10.7717/peerj.19931_

## Round 0.1 · original submission · Major Revisions

Having read the manuscript myself, I believe that there are no major flaws leading to a desk rejection. However, I do agree with R1 regarding concerns on the data analysis. Although I think that your approach of using ROC analyses is interesting, as it is now is very difficult to understand and the reader cannot understand why you followed such a path for your analyses. You need to better explain the rationale behind this decision, and also you need to give some context to the ROC analysis (trivially, you should explain what they and for what they are used, without assuming all readers are familiar with such procedures). In general, the statistical analysis should be better described. Moreover, I agree with R1 on the fact that the method section is not well explained, and, as it is, it does not allow for the replicability of the study.

Finally, at line 132-133, there is this sentence which is quite disturbing: "
Pearson correlation analysis was used between three dependent and four independent variables to assess the significance of the relationships."

First of all, correlations are just a standardized measure of the covariance between variables, there are no dependent and independent variables. Plus, at best, the correlation coefficient can be interpreted as an effect size measure, so I don’t understand how the significance of the relationship was tested.

·

Basic reporting

Abstract
The abstract left me feeling confused about the rationale of the study and how this aligns with the data analysis recommended. I would recommend a much simpler data analysis design – correlations and Hierarchical Multiple Linear Regression for this study. What has been done seems convoluted and complex in comparison.
Literature Review
The introduction and literature review does not do a good job of building a rationale for the study and the data analysis that has been selected. It should be written in the following format
Paragraph1 & 2 – Why this topic is important (quality of life and mental distress in university students) citing recent statistics for the prevalence in this population.
Paragraph 3 – define key terms – quality of life, mental distress, how are ultra-processed foods defined and how these are considered in this manuscript
Paragraphs 4 to 7 – What is currently known about this topic – introducing recent nutritional psychiatry literature on diet and mental health – the manuscript is missing key papers on UPF and mental health from this field.
Paragraphs 8 & 9 – What is not currently known in the literature -what is the gap that this manuscript will fill building a rationale for the data analysis applied to the study.
Paragraph 10 – What is the aim of this study? What are the research questions and hypotheses. These are needed to map on to the data analysis strategy that comes later.

Experimental design

The methods are not written in such a way that another researcher could replicate exactly what was done and achieve a similar results. Key information is missing such as participant recruitment, identification of the study design and what the key outcome variables, predictor variables and covariates are. Details of data-cleaning and power analysis are missing. Procedure also missing – what did the participants experience from the time they were recruited into the study until they were debriefed out of the study? Reliability and validity of the pre-validated scales is needed. Most importantly the data analysis strategy is convoluted and complex and it is unclear why this type of analysis has been applied to this study – I recommend simplifying to correlations and HMLR. There is a mention of qualitative data when I think you mean categorical data for n and %. The way the variables have been created PCS, MDS is confusing and they appear out of nowhere. Why have they been created like this?

Validity of the findings

More information needed in text for participant characteristics. Report the interesting information about all the variables to give the reader more of an idea of what the participants look like as a group.
Figures and tables are reported incorrectly. A table for a correlation matrix would describe this data in better detail than a figure with all the colours. It is just confusing.
Need to report the HMLR in more detail. How were the blocks ordered, how much variance in each block of the model. How much unique variance does the predictor variable have after adjusting for covariates in the model.
The ROC analysis has no rationale, why was this performed? It is confusing and I do not think it adds to the manuscript.

Additional comments

The data for this manuscript should be reconsidered and reanalysed. As it is written there is not enough detail for another researcher to replicate the findings and the analysis is convoluted and complex and does not make sense.

Reviewer 2 ·

Basic reporting

Abstract:
• Line 19: The researchers should begin by introducing what UPF (Ultra-Processed Foods) is, rather than mentioning the limited studies revealing its effects on quality of life and mood.
• Line 22: Remove "67.2% of women."
• Line 23: Replace "a survey form" with "a questionnaire" and specify whether the questionnaire was distributed face-to-face or online. The location of the study is also not mentioned and should be included.

Introduction:
• Lines 41–42: A citation should be provided.
• Line 42: More detailed information about the NOVA classification is needed.
• Line 47: The citations should follow the word "cancer."
• Line 49: Provide specific amounts of UPF consumption associated with the mentioned phenomenon.
• Line 62: Elaborate further on the limited access to nutritious, homemade meals among students.
• Line 63: It is inappropriate to state that university canteens offer easily accessible, fast-consumable options—this claim needs rephrasing or justification.
• The problem statement is not well-justified and requires better framing.

Experimental design

Methods:
• Line 77: Replace "&" with "and."
• Line 81: Rewrite "Then …applied" for better clarity.
• Provide the inclusion and exclusion criteria for the study, as well as the sample size calculation.
• Clarify if the questionnaire was self-administered by the respondents.
• Line 83: Why were extreme values removed? Provide a citation or justification for this decision.
• Line 80: Append the questionnaire used in the study as a supplementary file.
• Report the Cronbach's alpha for the instruments used to measure quality of life and mental distress, to verify reliability.

Validity of the findings

Results:
• Table 1 is confusing. Separate the continuous data from the categorical data for clarity.

Additional comments

Discussion:
• Line 185: physical?
• A recommendation for further studies is not provided and should be included.

Conclusion
The conclusion segment looks fine.

---

## Round 0.2 · Minor Revisions

Dear Authors,

I agree with R1 concerning the use of AI for writing the manuscript (which you do acknowledge at the end). The contrast between the AI generated section of the manuscript and the non-AI sections (e.g., methods) is quite noticeable. I do not condemn the use of AI as a supporting tool for writing, but in this case it feels like the entire introduction has been AI-generated. Please, consider re-writing it.
I see that you added some context to the ROC analysis, but It still think it is not enough to justify their use to the readers, and they are not quite well explained. If you cannot provide a clear rationale of the reasons why you decided to add such approach (besides “ROC approach can be used in nutrition and health science to identify an optimum cut-off point (Rankinen et al., 1999; Mirmiran et al. 2004”) as well as a clear explanation of the approach, I am afraid that I have to follow the suggestion of R1 of removing such analyses from the manuscript.

·

Basic reporting

I have high suspicion that the newly revised introduction may have been written by Generative AI software. The language proficiency in the introduction and in the methods are entirely different. Additionally elements of revision have been completed at surface-level and not clearly explained (e.g. how sample size was obtained).

Experimental design

I am still not convinced that the ROC was the correct analysis for this study OR it is still not explained well enough that readers would understand the data analysis strategy and its rationale.

Validity of the findings

The complexity of the ROC may be covering for non-sig or weak findings. I would not recommend for publication including this analysis.

Additional comments

My original hesitations remain regarding this manuscript. I would not recommend for publication.

Reviewer 2 ·

Basic reporting

The authors have amended the manuscript as per the reviewers' suggestions and, hence, it can be accepted in its present form.

Experimental design

The authors have amended the manuscript as per the reviewers' suggestions and, hence, it can be accepted in its present form.

Validity of the findings

The authors have amended the manuscript as per the reviewers' suggestions and, hence, it can be accepted in its present form.

---

## Round 0.3 · Minor Revisions

Could you just please add the same rationale and example for the use of ROC in the manuscript as you did in the response to me and to the reviewers? I was just asking you to provide a rationale and a wider framework for the use of this analysis, and you did that brillantly in the response letter. Why can't you do that in the manuscript?

---

## Round 0.4 · Minor Revisions

First, I apologize for the long delay in responding to your most recent submission. Given that the prior reviewers had extremely different opinions on the manuscript, it was necessary to obtain a third review to help mediate between these extremes. Unfortunately, it took a very long time to identify a qualified reviewer; however, we now have a high-quality third review in hand.

The new review is in general quite positive, but identifies some issues that need attention. In my opinion these qualify as minor revisions, which I hope you will be able to undertake.

In your last response to reviewers, you provided a rationale for use of ROC analysis; however, it appears that the information is only in the response letter, but should also be provided in the paper itself. Please consider adding that information to the manuscript (this overlaps with a comment by Reviewer 3).

Lastly, I need to raise several issues regarding statistical analysis:

1. In reporting ROC analysis, please note that an AUC>0.5 does *not* represent "compelling evidence for substantial discrimination" (line 236); in fact, given equal base rate of outcomes, 0.5 is chance performance, and minimally acceptable criteria for predictive power normally fall in the range from 0.7-0.8. If you are making a case for a different criterion, please justify and provide supporting citations.

2. In lines 178-179, it appears that 12 correlation analyses were run; it is unclear whether Bonferroni correction or similar method was used to protect significance.

3. In reporting the results of regression models (Table 2), please provide beta/p-values for all the predictors (in each model), not just for UPF.

4. Please provide means/SD and/or effect size (presumably obtained from Ruiz-Rofo et al., 2020) that you used to support your power analysis.

Please note that a revised version may need to return for re-review; however, again, it is my opinion that the comments raised by Reviewer 3 should be addressable, and I look forward to receiving a revised version.

·

Basic reporting

.

Experimental design

.

Validity of the findings

.

Additional comments

.

·

Basic reporting

The article presents clear and unambiguous language throughout, with professional and consistent use of English. The literature review is comprehensive, with sufficient references to relevant studies, and the background/context is well-established. The structure adheres to the required standards, and the article is professionally formatted with appropriate tables and figures. The raw data was shared, as per PeerJ's policies. However, the clarity of some figures and tables could be enhanced, particularly regarding the detailed description of the data. In addition, while the methodology is described with sufficient detail, more specific information about the statistical models used could be included for clarity.

Experimental design

he experimental design of this study is well-structured and clearly outlined. However, one area for improvement is the sample selection. While the study includes 595 participants, there is no discussion about how potential biases in participant recruitment were controlled for. It would be beneficial to provide more detail on the inclusion and exclusion criteria to ensure that the sample is representative of the university student population.

Additionally, although the study uses linear regression and ROC analysis to assess relationships and thresholds, more justification could be given for the choice of these statistical methods. It might be helpful to explain why these particular models were chosen over other possible analytical approaches and whether any model assumptions were tested. Providing more information on the assumptions behind these models and the potential limitations would strengthen the transparency and reproducibility of the results.

Lastly, while the study makes a strong case for the relationship between UPF consumption and mental health, the cross-sectional nature of the design limits the ability to draw causal conclusions. Suggesting future research using longitudinal or experimental designs would help in addressing this limitation.

Validity of the findings

The findings of the study are statistically sound, with robust data provided for analysis. The use of linear regression and ROC analysis offers solid support for the conclusions drawn about the relationship between ultra-processed food (UPF) consumption and mental health. However, a key area for improvement is the lack of an explicit discussion on the impact and novelty of the findings. While the study makes a valuable contribution to understanding the effects of UPFs on quality of life and mental distress, it would be helpful to further emphasize how these findings add to the existing body of literature, especially given the increasing global consumption of UPFs. A more thorough comparison with previous studies and highlighting the novel aspects of this research would strengthen the paper's contribution.

Additionally, while all underlying data has been provided, a more detailed explanation of how the data was controlled for potential confounding variables (beyond age, sex, and physical activity) could be included. This would help ensure the transparency of the statistical analysis and increase the trustworthiness of the results. Although the conclusions are generally well-stated and directly linked to the research question, it would be beneficial to briefly discuss the limitations of the study in relation to data interpretation, particularly given the cross-sectional design.

Lastly, while the conclusions are supported by the results, a more explicit statement about the potential real-world applications of these findings, particularly in terms of public health recommendations, would strengthen the conclusion.

Additional comments

Overall, the manuscript provides valuable insights into the relationship between ultra-processed food (UPF) consumption and mental health in university students. The study's clear structure and well-defined methodology make it a significant contribution to the field of nutrition and mental health. However, there are a few areas where further clarification could enhance the manuscript.

Firstly, while the study uses the NOVA classification system to categorize food groups, more information on the cultural context of food choices in the specific population under study would be helpful. Given that the research is conducted in Türkiye, some discussion on how local dietary patterns might influence UPF consumption and the generalizability of these findings to other populations would be beneficial.

Additionally, the manuscript could benefit from a more thorough discussion of potential interventions to mitigate the negative effects of UPF consumption on physical and mental health. While the authors mention the need for strategies to decrease UPF intake, specific policy recommendations or suggestions for universities to promote healthier dietary habits could provide actionable insights for stakeholders.

Finally, there are a few minor language and formatting issues that could be addressed. A careful proofreading of the manuscript would improve the overall clarity and flow of the text. Additionally, some figures, such as those in the tables and ROC analysis, could benefit from more detailed captions to ensure they are fully interpretable without needing to refer to the main text.

---

## Round 0.5 · accepted · Accept

I have read the revised version carefully, and it is my opinion that previous reviewer comments have been adequately addressed, and that the manuscript is now ready for publication.